# Incomplete antiviral treatment may induce longer durations of viral shedding during SARS-CoV-2 infection

Kwang Su Kim[1],*, Shoya Iwanami[1],*, Takafumi Oda[2], Yasuhisa Fujita[1], Keiji Kuba[3], Taiga Miyazaki[4], Keisuke Ejima[5],*, Shingo Iwami[1,6,7,8,9,10],*

**The duration of viral shedding is determined by a balance between de novo infection and removal of infected cells. That is, if infection is completely blocked with antiviral drugs (100% inhibition), the duration of viral shedding is minimal and is determined by the length of virus production. However, some mathematical models predict that if infected individuals are treated with antiviral drugs with efficacy below 100%, viral shedding may last longer than without treatment because further de novo infections are driven by entry of the virus into partially protected, uninfected cells at a slower rate. Using a simple mathematical model, we quantified SARS-CoV-2 infection dynamics in non-human primates and characterized the kinetics of viral shedding. We counterintuitively found that treatments initiated early, such as 0.5 d after virus inoculation, with intermediate to relatively high efficacy (30–70% inhibition of virus replication) yield a prolonged duration of viral shedding (by about 6.0 d) compared with no treatment.**

## Introduction

Two main processes are involved in the pathogenesis of coronavirus disease 2019 (COVID-19): the disease primarily develops through replication of severe acute respiratory syndrome coronavirus 2 (SARS-CoV-2) and is then driven by exaggerated host immune/inflammatory responses to the virus, leading to various tissue damage. Thus, antiviral agents are considered to likely be more beneficial in the earlier stages of COVID-19, whereas immunosuppressive/anti-inflammatory therapies may have the greatest effects in later stages of the disease (1).

Remdesivir (RDV) is currently the only antiviral drug approved by the Food and Drug Administration for the treatment of COVID-19 and is recommended for hospitalized patients who require supplemental oxygen. However, it is not routinely recommended for patients at advanced stages of the disease, such as for patients maintained on artificial respiration by means of mechanical ventilators, owing to the lack of data showing benefit (2, 3, 4, 5). The corticosteroid dexamethasone has been reported to improve survival in hospitalized patients who require supplemental oxygen, especially in patients who require mechanical ventilation (6, 7, 8, 9).

Although mortality is an important and ultimate clinical outcome at both the individual and the population level (10), the efficacy of antiviral drugs that reduce infection with the virus or viral replication may also be evaluated primarily by using viral load kinetics (5, 11, 12, 13). Although viral load was similar among symptomatic and asymptomatic outpatients in a previous study (14), high SARS-CoV-2 viral load on admission is associated with in-hospital mortality in COVID-19 patients who require hospitalization (15, 16). In addition, reporting viral load has an important role in infection prevention practices, given that viral load correlates with levels of infectiousness (17, 18, 19), even though detection of viral RNA by PCR does not necessarily indicate the infectivity of SARS-CoV-2 in the late phase of disease.

The currently available data suggest that reducing the amount of viral load and/or the duration of viral shedding might be beneficial for suppressing early inflammation in patients or for preventing further human-to-human transmission (20). An effective antiviral drug for COVID-19 is urgently needed, and the duration of viral shedding would be an important clinical outcome for assessing drug efficacy. Indeed, to date, several clinical studies have used the duration of viral shedding as an outcome to evaluate the efficacy of candidate drugs (11, 13).

In general, antiviral drugs that have limited efficacy for inhibiting viral replication are not expected to substantially affect virus infection dynamics in the clinical setting regardless of merit. Exhaustive simulations of mathematical models for antiviral treatment; however, sometimes uncover unnatural behaviors even under realistic parameter ranges. For example, it is observed that viral shedding may

[1]Interdisciplinary Biology Laboratory (iBLab), Division of Biological Science, Graduate School of Science, Nagoya University, Nagoya, Japan   [2]Department of Biology, Faculty of Sciences, Kyushu University, Fukuoka, Japan   [3]Department of Biochemistry and Metabolic Science, Akita University Graduate School of Medicine, Akita, Japan   [4]Department of Infectious Diseases, Nagasaki University Graduate School of Biomedical Sciences, Nagasaki, Japan   [5]Department of Epidemiology and Biostatistics, Indiana University School of Public Health-Bloomington, Bloomington, IN, USA   [6]Institute of Mathematics for Industry, Kyushu University, Fukuoka, Japan   [7]Institute for the Advanced Study of Human Biology (ASHBi), Kyoto University, Kyoto, Japan   [8]NEXT-Ganken Program, Japanese Foundation for Cancer Research, Tokyo, Japan   [9]Interdisciplinary Theoretical and Mathematical Sciences Program (iTHEMS), RIKEN, Saitama, Japan   [10]Science Groove Inc., Fukuoka, Japan

Correspondence: kejima@iu.edu; Correspondence: iwami.iblab@bio.nagoya-u.ac.jp
*Kwang Su Kim and Shoya Iwanami contributed equally to this work

**Table 1. Estimated parameters (fixed effect) for SARS-CoV-2 infection in nose and throat.**

| Parameter name | Symbol (unit) | Nose | Throat |
|---|---|---|---|
| Parameters in the model | | | |
| Maximum rate constant for viral replication | $\gamma$ (d$^{-1}$) | 18.2 | 2.89[a] |
| Rate constant for virus infection | $\beta$ ([copies/ml]$^{-1}$ d$^{-1}$) | $1.79 \times 10^{-6}$ | $6.70 \times 10^{-6}$ |
| Death rate of infected cells | $\delta$ (d$^{-1}$) | 1.14 | |
| Efficacy of blocking virus production by RDV | $\varepsilon$ (–) | 0.618 | |
| Viral load at virus inoculation | $V(0)$ (copies/ml) | $2.86 \times 10^{3}$ | |
| Quantities derived from the parameters | | | |
| Within-host basic reproduction number | $R_0$ (=$\gamma/\delta$) | 16.0 | 2.54[a] |
| Malthusian parameter | $M$ (=$\gamma - \delta$) | 17.1 | 1.75[a] |

[a]Statistically different from nose (the Wald test).

paradoxically last longer with treatment than without treatment if the antiviral efficacy is below 100% ([21], [22]). Although even simple limited models of target cells with antiviral treatment blocking virus production (or virus entry) can explain this paradoxical phenomenon, it has not yet been reported in both a clinical and an experimental setting. In a recent article ([23]), these paradoxical phenomena were observed for the first time in a SARS-CoV-2 infection non-human primate model treated with RDV, which is known as a nucleoside analogue. In the present study, we used a simple mathematical model and its extended models to quantify SARS-CoV-2 infection dynamics in the non-human primate and to characterize viral shedding to better understand the viral infection dynamics behind the paradoxically longer duration of viral shedding with antiviral treatment.

## Results

### Characterizing SARS-CoV-2 infection dynamics in different specimens from rhesus macaque

We quantitatively analyzed time-course data on viral load from nose and throat swabs of rhesus macaques infected with SARS-CoV-2 (reported in references [23] and [24]) using a viral dynamics model (see the Materials and Methods section). To consider interindividual variations in viral loads, we used a nonlinear mixed-effect modelling approach to estimate parameters (see the Materials and Methods section). The estimated fixed effect parameters for nose and throat swabs are listed in Table 1, and the individual estimated parameters for each macaque are listed in Table S1. When we compared the population parameters between specimens, the maximum rate constant for viral replication was statistically significantly larger for nose swabs than for throat swabs ($P = 4.12 \times 10^{-10}$ by the Wald test). The viral load curves based on the individual parameters from the rhesus macaques are depicted for nose (curves in purple) and throat (curves in orange) swabs in Fig 1A.

Expected viral loads in nose and throat swabs (with estimated fixed effect parameters) are shown in Fig 1B and C, respectively. The viral RNA load in nose swabs peaked earlier than in throat swabs: at

0.680 d after virus inoculation compared with 3.09 d after inoculation in throat swabs. The peak timing in the nose was numerically earlier than in the throat, although the difference among specimens was not significant. We further calculated and compared the following quantities, which are composed of the estimated parameters (Table 1): the within-host basic reproduction number ($R_0 = \gamma/\delta$), which is the average number of newly infected cells produced by any single infected cell when target cells are fully uninfected ([22]), and the Malthusian parameter ($M = \gamma - \delta$), which is an indicator of the initial growth rate of the viral load ([25], [26]). Both $R_0$ and $M$ of SARS-CoV-2 were significantly larger in nose swabs than in throat swabs ($P = 1.98 \times 10^{-10}$ and $1.54 \times 10^{-11}$ by the Wald test, respectively) (Table 1). Thus, both the larger $R_0$ and $M$ and the earlier peak viral load in nose swabs suggest that the virus replicates and spreads more effectively in the nose than in the throat, implying that the infection dynamics of SARS-CoV-2 differs in different organs. These organ-specific infection dynamics may thus affect treatment outcomes in different organs. In the next section, we provide a detailed analysis of anti-SARS-CoV-2 treatment with RDV in infected rhesus macaques.

### Evaluation of antiviral treatment with remdesivir in SARS-CoV-2–infected rhesus macaque

In addition to our finding of the organ specificity of virus infection dynamics, other differences between organs may affect drug efficacy, such as target cell constitutions, drug distributions in plasma and tissue compartments, etc. ([27], [28]). To quantitatively evaluate how these organ-specific heterogeneities affect treatment outcomes, we further analyzed time-course data on viral load from the same two specimens of infected rhesus macaques treated with RDV. Note that the study of RDV-treated rhesus macaque reported that there was no significant reduction in viral load in the nose or throat at each time point although RDV suppressed virus infection in the lower respiratory tract ([23]). Here we used the above parameter values obtained from the SARS-CoV-2–infected rhesus macaques without treatment, and then estimated the antiviral efficacy of RDV against SARS-CoV-2 in the nose and throat (Table 1). The viral load curves are depicted with observed data in Fig 1D for nose and throat swabs.

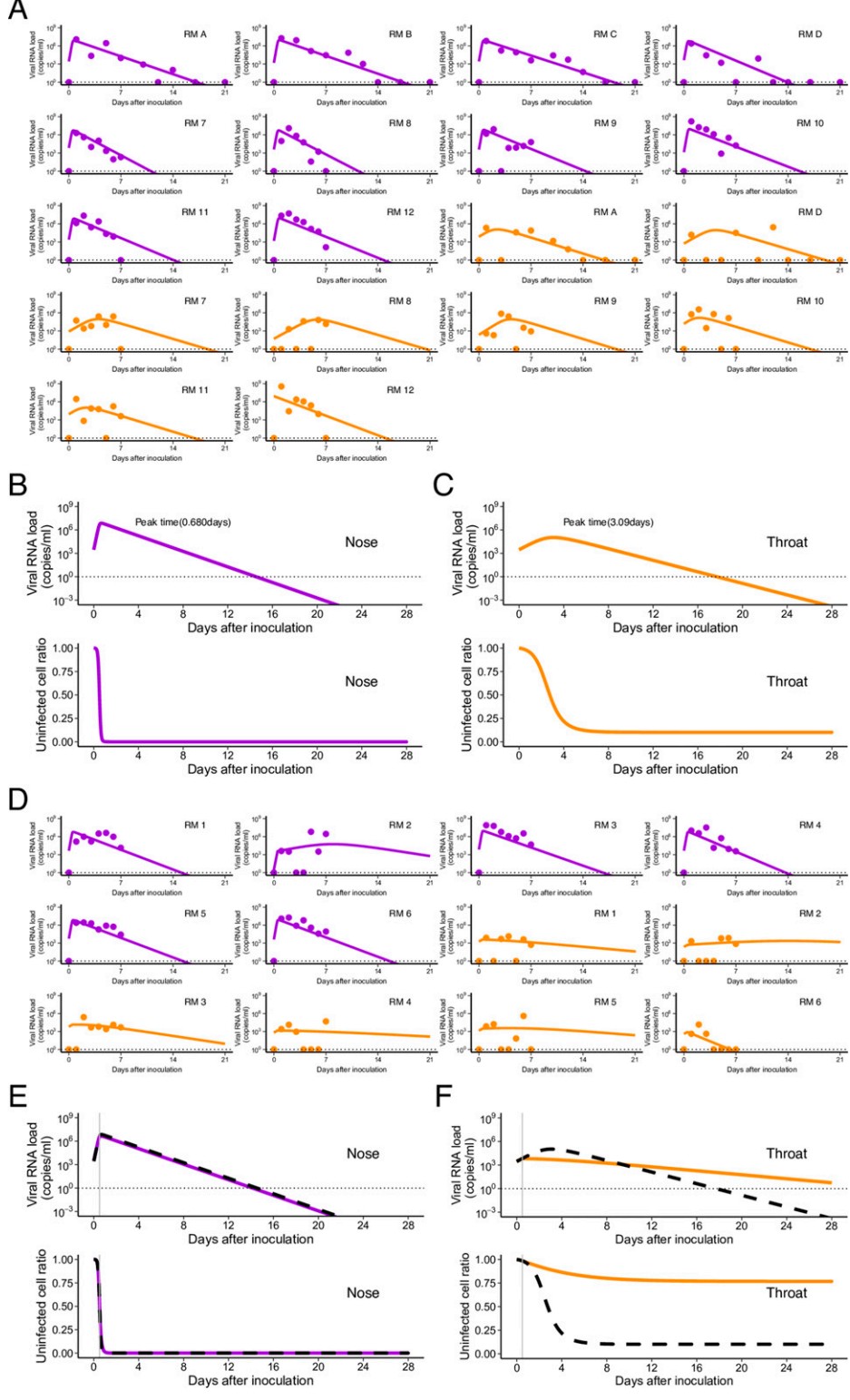

**Figure 1. Comparison of viral load trajectory between nose and throat swabs in SARS-CoV-2–infected rhesus macaques without and with treatment.**
Viral loads were measured using nasal (purple) and throat (orange) swabs for SARS-CoV-2–infected macaques. **(A)** The closed dots and curves correspond to the observed and estimated viral loads in the macaques without treatment (A). **(B, C)** The expected viral load and uninfected target cell proportion in nose, throat, and lung are calculated by the population parameter and shown in (B, C) (without treatment). **(D)** and (E, F) are same as (A) and (B, C), respectively, but for the group with treatment. **(B, C, E, F)** The black dashed curves in (E, F) represent the expected viral load and uninfected target cell proportion shown in (B, C). The black horizontal dotted and gray vertical lines show the detection limit of viral load and the timing of initiation of the treatments (0.5 d), respectively. Source data are available online for this figure.

The fixed effect of the antiviral efficacy in throat is estimated to be ε =0.618 (95% CI: 0.373–0.863), although we failed to estimate individual antiviral efficacy in nose (see below). This is because the treatment with RDV was initiated at day 0.5, which was close to the peak viral load in nose (i.e., day 0.680; Fig 1B). Because only a very small fraction of target cells remain uninfected after the viral load peak (29, 30), the number of ongoing de novo infections that the RDV can potentially interrupt is limited. Therefore, even if RDV blocked

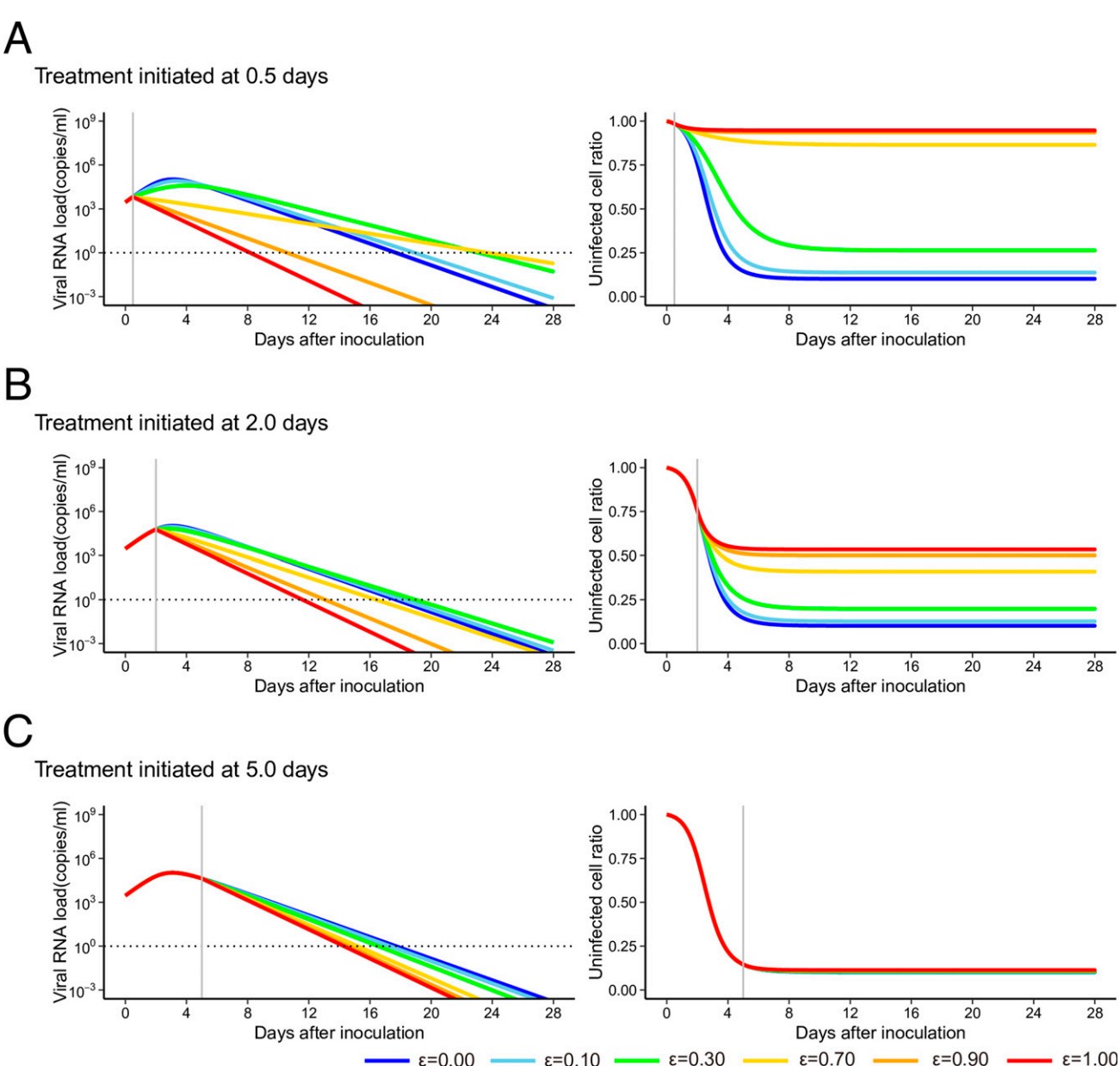

**Figure 2. Different treatment outcomes in throat as the result of varying drug efficacy and the timing of initiation.**
**(A, B, C)** The expected viral load and uninfected target cell proportion are calculated under the condition of antiviral treatment blocking virus production with different efficacies (ε = 0.00, 0.10, 0.30, 0.70, 0.90, and 1.00, corresponding to the blue, light blue, green, yellow, orange, and red solid lines, respectively) and with initiation at 0.5, 2.0, or 5.0 d after inoculation using the estimated model parameter values for throat shown in (A, B, C), respectively. The black horizontal dotted and gray vertical lines show the detection limit of viral load and the timing of initiation of the treatments, respectively.
Source data are available online for this figure.

virus production with relatively high efficacy up to 100%, the viral load decay was not significantly influenced (Figs 1E and S1A). We also quantitatively analyzed time-course data on viral load from bronchoalveolar lavage (BAL) samples (Fig S2A–D). The viral load decay was not influenced by treatment (Fig S2D) owing to the late initiation of RDV therapy (i.e., day 0.5), which was after the viral load peak in BAL fluid (i.e., 0.360 d; Fig S1B). In fact, there was little difference in the duration of viral shedding between rhesus macaques without and with RDV treatment (Figs 1E and S2D). On the other hand, because the initiation of RDV (i.e., day 0.5) was before the viral load peak in throat

(i.e., day 3.09; Fig 1C), blocking virus production affected the viral load and the duration of viral shedding (Fig 1F). We further investigated the treatment effect of varying the treatment efficacy and the timing of treatment initiation.

## Paradoxically longer durations of viral shedding with antiviral treatment

In our calculations using the nose data (see Fig 1E), the duration of viral shedding was not influenced much by treatment because

ongoing de novo infection was limited when treatment was initiated. By contrast, in the throat, the duration of viral shedding was determined by a balance between de novo infection and removal of infected cells. Surprisingly, as shown in Fig 1F, we found a trend for viral shedding to last longer in the macaques treated with RDV than in those without treatment despite more than 60% of de novo infection being blocked.

To elucidate a viral infection dynamics behind this counterintuitive longer duration of viral shedding in the throat, we further analyzed the target cell limited model for throat by varying drug efficacy and the timing of treatment initiation (Fig 2). If the efficacy of treatment for blocking virus production was close to 100% (Fig 2A and B), the duration of viral shedding was shortened because all ongoing de novo infections were blocked. However, with an intermediate efficacy of treatment (10–70%), the duration became longer because uninfected cells were temporally protected by the treatment but some ongoing de novo infections were not blocked. Thus, the protected cells were gradually infected and drove further but slower infection than in the case without treatment (Fig 2A and B). For low-efficacy treatments (<10%), the treatments no longer protected uninfected cells and de novo infections occurred at almost same level without treatment (i.e., the duration became shorter again) (Fig 2A and B). This trend was highlighted when the treatment was initiated early. As we discussed above, limited uninfected target cells remained after the viral load hit its peak. Thus, the treatment impact on the duration of viral shedding was limited when the treatment was initiated after the viral load peak (Fig 2C, see also Fig 1E). We further compared the model we used in this study and other extended models, all of which have been used to describe virus dynamics of SARS-CoV-2 and other viruses, to show our conclusion is consistent without loss of generality (see Figs S3A–H, S4A–H, and S5A–H).

Taken together, under incomplete antiviral treatments, our findings show that uninfected target cells remain longer and infection slowly continues depending on target cell availability. The availability of target cells might explain the paradoxical phenomenon in different organs first observed in the study of SARS-CoV-2 infection in the non-human primate model.

## Discussion

It is well known that entry of SARS-CoV-2 into cells requires interaction of the virus spike protein with host angiotensin-converting enzyme 2 (ACE2) receptor, and cleavage and activation of spike by a serine protease, TMPRSS2 (31). Analyses of single-cell sequencing datasets have demonstrated that ACE2 receptor is abundantly expressed in multiple organs, such as nose, lung, eye, and intestine (27), which suggests that the target organ of SARS-CoV-2 is not limited to respiratory tracts. Given that the expression level of ACE2 varies in different organs, virus infection dynamics is expected to differ as well. In fact, although the differences might be due to the sensitivity of the quantification methods used, different viral loads of SARS-CoV-2 in different specimens have been reported (32, 33, 34, 35). Whereas interferon signalling upon SARS-CoV-2 infection is suggested to induce ACE2 expression in single-cell levels (27), cellular entry of SARS coronavirus leads to down-regulation of bulk amounts of ACE2

protein in the lungs and heart (36, 37), which suggests a complicated mechanism for virus spread in the tissue microenvironment during the course of de novo infection of SARS-CoV-2. Thus, different dynamics of the virus and regulation of host factors in the organs might contribute to a paradoxically prolonged duration of virus shedding with incomplete RDV treatment.

In this study, we quantitatively analyzed time-course data on viral load from different specimens, such as nasal swabs, throat swabs, and BAL fluid, of SARS-CoV-2–infected rhesus macaques without and with RDV treatment (23, 24). Because direct and multi-route inoculations with a high viral titer were used in the infection experiments in the rhesus macaques, we observed extremely rapid infections in multiple organs compared with natural infections in humans via respiratory transmission. However, in these infection experiments, because the multi-route inoculations led to effective and rapid infections across organs, we found that the peak viral load appeared faster and that $R_0$ was larger in the nose than in the throat (Fig 1), implying different treatment outcomes in different organs. Because the initiation of the RDV treatment in the nose and lung (i.e., in BAL fluid) was close to the peak of viral load, the RDV had little effect on altering the viral load decay even though it completely blocked virus production; therefore, we could not observe an antiviral effect on the duration of virus shedding (Figs 1E and S2D). In contrast, interestingly, we demonstrated that in the throat an efficacy of RDV of more than 60% induced longer durations of viral shedding (Fig 1F). This paradoxically longer duration could be explained by our simple target cell limited mathematical model: because an incomplete antiviral treatment reduces ongoing infections and protects uninfected target cells, progeny virus that escape from treatment further induce de novo infections into protected, uninfected cells but at a slower rate (Fig 2). We quantitatively revealed the effect of RDV on SARS-CoV-2 replication in throat in treated rhesus macaque and explained why no clear effect of treatment in nose and lung is observed by using a mathematical model.

In many clinical trials, the duration of viral shedding has been used as a primary outcome to evaluate the antiviral effect of treatments with lopinavir and ritonavir (11), hydroxychloroquine (38, 39), and meplazumab (40). Because the viral load of SARS-CoV-2 peaks on or before symptom onset in many patients (18), and the mean interval between symptom onset and hospitalization is estimated to be 4.64 d (41), it is expected that antiviral treatments initiated after symptom onset will not shorten the duration of viral shedding. However, prophylactic use of antiviral drugs or contact-tracing-based (i.e., identifying patients before symptom onset) treatments may shorten the duration of viral shedding.

There are various studies using mathematical models that can explain SARS-CoV-2 infection dynamics and give insight into treatment strategy (29, 42, 43, 44). Another mathematical study of nasal viral load also showed a longer duration of virus shedding under RDV treatment; however, that study did not directly estimate treatment efficacy (45). Moreover, the previous discussion of the difference in SARS-CoV-2 viral dynamics in lung and nose used a more detailed mathematical model (46 *Preprint*). Because our mathematical model (two-dimensional ordinary differential equations, or ODEs) is derived from a basic virus dynamics model (three-dimensional ODEs [21]) by assuming quasi-steady state, we can reduce the

number of parameters needing to be estimated. Note that there might be a discrepancy in predicted virus infection dynamics between the two-dimensional and the three-dimensional ODEs if the antiviral efficacy is close to 100%. However, because the estimated antiviral efficacy is around 60%, the two-dimensional ODE well approximates the virus infection dynamics for the macaques treated with RDV. Ours is the first study to reveal the treatment conditions that induce longer durations of viral shedding based on data collected during treatment.

The limited number of rhesus macaques in the SARS-CoV-2 infection experiments (28 and 17 animals with and without treatment, respectively, in references 23 and 24) used in our analysis might be a limitation. For example, we could not obtain statistically significant differences for the time from virus inoculation to the viral load peak between nose and throat swabs because of large variance due to the small sample size. Once more experimental datasets become available, our mathematical model may be able to more precisely differentiate the virus infection dynamics of different organs. In addition, although relevant data and evidence for prophylactic use of antiviral drugs for SARS-CoV-2 infection in humans are not available so far, future studies should verify our findings by using human data. Thus, collecting longitudinal data and quantifying the viral dynamics with mathematical models are important to evaluate outcomes of antiviral treatments.

## Materials and Methods

### Study data

The longitudinal viral load was extracted from the published studies of SARS-CoV-2 infection in a non-human primate model (23). The rhesus macaques were inoculated with a total dose of $2.6 \times 10^6$ TCID$_{50}$ through intranasal, oral, ocular, and intratracheal routes. After the inoculations, the viral load was measured every day. Also, BAL was performed using 10 ml of sterile saline to extract viral RNA in BAL fluid. The rhesus macaques were assigned to receive either an intravenous dose of RDV (10 mg/kg loading dose on 0.5 d after inoculation with SARS-CoV-2, followed by 5 mg/kg daily) or an equal volume of vehicle solution (2 ml/kg loading dose, followed by 1 ml/kg daily). We also added viral loads from nose and throat of rhesus macaques infected with SARS-CoV-2 with the same experimental protocol in (24). Data from macaques with less than three data points above the detection limit were excluded. Finally, we used data from 28 and 17 animals without and with treatment, respectively.

### Mathematical model

The following mathematical model describes SARS-CoV-2 viral dynamics (29, 30, 47):

$$\frac{df(t)}{dt} = -\beta f(t)V(t), \tag{1}$$

$$\frac{dV(t)}{dt} = (1 - \varepsilon H(t))\gamma f(t)V(t) - \delta V(t). \tag{2}$$

The variables $f(t)$ and $V(t)$ are the relative fraction of uninfected target cells at time $t$ to those at time 0, and the amount of virus at time, respectively. The parameters $\beta$, $\gamma$, and $\delta$ represent the rate constant for virus infection, the maximum rate constant for viral replication, and the death rate of virus-producing cells, respectively. To describe the off- and on-treatments, we employed a Heaviside function: $H(t) = 0$ if $t < t^*$ (i.e., before treatment initiation); otherwise $H(t) = 1$. $\varepsilon$ is the efficacy of antiviral treatment blocking virus production ($0 < \varepsilon \le 1$). If $\varepsilon = 1$, the virus replication from the infected cells is perfectly inhibited (i.e., the antiviral efficacy is 100%). We evaluated the expected antiviral effect of the treatment on the duration of virus shedding under different inhibition rates ($\varepsilon$) and initiation timings ($t^*$) [Equations (1) and (2)].

### The nonlinear mixed-effect model

All viral load data were fitted using a nonlinear mixed-effect modelling approach, which estimates population parameters while accounting for interindividual variation in virus dynamics. The model included both a fixed effect (constant across rhesus macaques) and a random effect (different between rhesus macaques) for each parameter. The parameter values for rhesus macaque $i$ can be expressed as $\vartheta_i (= \vartheta \times e^{\pi_i})$, where $\vartheta$ is a fixed effect and $e^{\pi_i}$ is a random effect; $\pi_i$ is normally distributed as $N(0, \Omega)$. We estimated fixed effects and random effects using the stochastic approximation expectation-maximization algorithm and the empirical Bayes's method, respectively. In addition, we used location of specimens for measuring viral load as a categorical covariate in estimating $\gamma$ and $\beta$ which provide the lowest Bayesian information criterion. Fitting was carried out using MONOLIX 2019R2. To handle data under the detection limit, the likelihood was constructed assuming that the data are within the interval from 0 to $\alpha$, where $\alpha$ is the lowest observed viral load (above the detection limit) around the day when the viral load is censored (48). The viral load trajectories were depicted using the best fit parameter estimates for individual data in Figs 1 and S1. The estimated parameters (fixed effect and individual parameter) and initial values are summarized in Tables 1 and S1.

### Statistical test

To evaluate statistical differences for each of the estimated parameters ($\gamma$, $\beta$, $R_0$, and $M$), we applied the Wald test. The time from virus inoculation to viral load peak was calculated at the population level by running the model using estimated fixed effects and the initial values. To compare the timing of viral load peak in the nose, throat, and lung, the differences of these values were tested by the Jackknife test (49, 50).

## Supplementary Information

# Acknowledgements

This study was supported in part by the Basic Science Research Program through the National Research Foundation of Korea funded by the Ministry of Education 2019R1A6A3A12031316 (to KS Kim); Grants-in-Aid for JSPS Research Fellow 19J12319 (to S Iwanami); Scientific Research (KAKENHI) B 18KT0018 (to S Iwami), 18H01139 (to S Iwami), 16H04845 (to S Iwami), Scientific Research in Innovative Areas 20H05042 (to S Iwami), 19H04839 (to S Iwami), 18H05103 (to S Iwami); AMED CREST 19gm1310002 (to S Iwami); AMED Japan Program for Infectious Diseases Research and Infrastructure, 20wm0325007h0001, 20wm0325004s0201, 20wm0325012s0301, 20wm0325015s0301 (to S Iwami); AMED Research Program on HIV/AIDS 19fk0410023s0101 (to S Iwami); AMED Research Program on Emerging and Re-emerging Infectious Diseases 19fk0108050h0003 (to S Iwami), 19fk0108156h0001 (to S Iwami), 20fk0108140s0801 (to S Iwami) and 20fk0108413s0301 (to S Iwami); AMED Program for Basic and Clinical Research on Hepatitis 19fk0210036h0502 (to S Iwami); AMED Program on the Innovative Development and the Application of New Drugs for Hepatitis B 19fk0310114h0103 (to S Iwami); JST MIRAI (to S Iwami); Moonshot R&D grant Number JPMJMS2021 (to S Iwami) and JPMJMS2025 (to S Iwami); Mitsui Life Social Welfare Foundation (to S Iwami); Shin-Nihon of Advanced Medical Research (to S Iwami); Suzuken Memorial Foundation (to S Iwami); Life Science Foundation of Japan (to S Iwami); SECOM Science and Technology Foundation (to S Iwami); The Japan Prize Foundation (to S Iwami); Daiwa Securities Health Foundation (to S Iwami).

## Author Contributions

KS Kim: data curation, formal analysis, investigation, visualization, methodology, and writing—original draft, review, and editing.
S Iwanami: data curation, formal analysis, investigation, visualization, and methodology.
T Oda: data curation, formal analysis, validation, investigation, visualization, methodology, and writing—original draft, review, and editing.
Y Fujita: data curation, formal analysis, and methodology.
K Kuba: conceptualization, investigation, and writing—original draft, review, and editing.
T Miyazaki: conceptualization, investigation, and writing—original draft, review, and editing.
K Ejima: conceptualization, data curation, formal analysis, investigation, methodology, project administration, and writing—original draft, review, and editing.
S Iwami: conceptualization, supervision, funding acquisition, investigation, methodology, project administration, and writing—original draft, review, and editing.

## Conflict of Interest Statement

The authors declare that they have no conflict of interest.

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
