## [Reviewer comments · Life Science Alliance]

Life Science Alliance

Incomplete viral treatment may induce longer durations of viral shedding during SARS-CoV-2 infection

Kwang Su Kim, Shoya Iwanami, Takafumi Oda, Yasuhisa Fujita, Keiji Kuba, Taiga Miyazaki, Keisuke Ejima, and Shingo Iwami

DOI: <https://doi.org/10.26508/lsa.202101049>

Corresponding author(s): Shingo Iwami, Department of Biology, Faculty of Sciences, Kyushu University and Keisuke Ejima,

Review Timeline:

Submission Date:	2021-02-05
Editorial Decision:	2021-05-19
Revision Received:	2021-06-19
Editorial Decision:	2021-07-19
Revision Received:	2021-07-23
Accepted:	2021-07-23

Transaction Report:

May 19, 2021

Re: Life Science Alliance manuscript #LSA-2021-01049

Dr. Shingo Iwami
Department of Biology, Faculty of Sciences, Kyushu University
Motooka
Fukuoka 8190395
Japan

Dear Dr. Iwami,

Thank you for submitting your manuscript entitled "Incomplete antiviral treatment induce longer durations of viral shedding during SARS-CoV-2 infection" to Life Science Alliance. The manuscript was assessed by expert reviewers, whose comments are appended to this letter.

We apologize for this unusual and extended delay in getting back to you. We had trouble in finalizing a panel of reviewers for this study, and in the interest of avoiding any further delay, we have decided to return to you with the comments of one of the experts, who thinks that while the paper is well written, there are some novelty and technical concerns that should still be addressed.

This is a highly unusual case for Life Science Alliance, but given the overall positive outlook of the reviewer and in the interest of time, we encourage you to submit a revised version of the manuscript that addresses the reviewer's concerns. For a fair peer-review, we will most likely reach out to an additional reviewer to assess the revised manuscript. We will walk them through the situation, so that they are aware that they will be looking at an already revised manuscript. Please let me know if you have any questions.

Thank you for this interesting contribution to Life Science Alliance. We are looking forward to receiving your revised manuscript.

Sincerely,

Shachi Bhatt, Ph.D.
Executive Editor
Life Science Alliance
<http://www.lsajournal.org>
Tweet @SciBhatt @LSAJournal

- A letter addressing the reviewers' comments point by point.
- An editable version of the final text (.DOC or .DOCX) is needed for copyediting (no PDFs).
- High-resolution figure, supplementary figure and video files uploaded as individual files: See our detailed guidelines for preparing your production-ready images, <https://www.life-science-alliance.org/authors>
- Summary blurb (enter in submission system): A short text summarizing in a single sentence the study (max. 200 characters including spaces). This text is used in conjunction with the titles of papers, hence should be informative and complementary to the title and running title. It should describe the context and significance of the findings for a general readership; it should be written in the present tense and refer to the work in the third person. Author names should not be mentioned.

B. MANUSCRIPT ORGANIZATION AND FORMATTING:

Reviewer #1 (Comments to the Authors (Required)):

In this manuscript, Kim et al. employ mathematical models to analyse SARS-CoV-2 infection

dynamics in the non-human primates in the presence or absence of remdesivir (a nucleotide analogue prodrug). The main model prediction is that early antiviral treatment with moderate efficacy (~30-70%) leads to a prolonged duration of viral shedding compared to infection duration without treatment. While the paper is well written, there are significant issues regarding the novelty and the technical details that would need to be addressed.

Specific comments:

1) In their recent work in PLoS Biology (PMID: 33750978), the authors made similar predictions on SARS-CoV-2 dynamics during antiviral treatments: 'Note that viral shedding may last longer with treatment than without treatment if the antiviral efficacy is below 100% and initiated early.' Another modelling study (PMID: 33097472) from a different group predicted that early initiation of subpotent remdesivir treatment prolonged the infection duration. Furthermore, a recent modelling study (PMID: 32882638) analysed SARS-CoV-2 infection dynamics during remdesivir treatment in the non-human primates and predicted that remdesivir might lengthen SARS-CoV-2 infection. Therefore, the novelty of this study is unclear.

2) In their model (Eqs. [1-2]), when $\epsilon = 1$, the viral load decline is given by $dV/dt = -\delta \cdot V$. In the standard SARS-CoV-2 dynamics model, when $\epsilon = 1$, the viral load decline is given by $dV/dt = -c \cdot V$. Here, δ and c are the infected cell death rate constant and virion degradation rate constant, respectively. Perhaps the authors could clarify this potential discrepancy.

3) It appears the authors have made pseudosteady state assumption ($dV/dt = 0$) to arrive at Eqs. [1-2]. Drug treatment (e.g., remdesivir) destroys pretreatment pseudosteady state (given by $V = p/c$) and establishes a new pseudosteady state (given by $V = p(1-\epsilon)/c$), where I is the infected cells, and p is viral production rate per infected cell. It is unclear whether their model (Eqs. [1-2]) can capture the viral load changes caused by changes in pseudosteady states due to remdesivir treatment initiation. Regardless, it would help if the authors could compare model predictions with and without pseudosteady assumption in the supplementary.

4) The models in this manuscript does not explicitly consider innate and adaptive immune responses. I appreciate that estimating parameter values of more complex models with immune components might not be possible with the current data set. However, the authors must investigate (at least qualitatively) whether subpotent remdesivir treatment leads to prolonged infection duration even if immune responses are explicitly considered. For instance, the innate immune response has been suggested to render target cells refractory to infection (e.g., PMID: 32558354, PMID: 22761567). Could this prevent the slower rate of infection during subpotent remdesivir treatment and reduce infection duration?

Minor comments:

5) It would help if the authors could describe how they treated the data points falling below the detection limit of viral during data fitting. Were these data points ignored or set to zero during fitting?

6) On lines 261, the authors mention remdesivir dosing as, '10mg/kg loading dose on 0.5 days after inoculation with SARS-CoV-2, followed by 262 5mg/kg daily.' Should the authors consider the efficacy to vary with time instead of constant efficacy (e.g., PMID: 33097472)?

7) On line 75-82, the authors mention, 'For example, in the field of virus dynamics (21, 22), it is observed that viral shedding may paradoxically last longer with treatment than without treatment if the antiviral efficacy is below 100% ... In a recent paper (23), these paradoxical phenomena were

observed for the first time in a SARS-CoV-2 infection non-human primate model treated with RDV ...' It seems experiments in ref. (23) measure the viral load only up to day 7 post-infection. Therefore, it is unclear how the authors interpret and state that the infection duration is longer with remdesvir (RDV) than without treatment in these experiments.

8) References 24 and 38 are the same.

9) On line 131, is it reference 24 or is it 23?

Response to the reviewers' comments

Kwang Su Kim, Shoya Iwanami, Takafumi Oda, Yashuhisa Fujita, Keiji Kuba,
Taiga Miyazaki, Keisuke Ejima and Shingo Iwami

June 19, 2021

We would like to thank the reviewers for their thorough reading of the manuscript and their excellent comments and suggestions. In this letter, we present the referees' comments in black and provide our responses in red. All changes made in addressing the reviewers' comments have been incorporated in the revised version of the manuscript, and we believe that the manuscript has been considerably improved by these changes.

Reviewer #1:

In this manuscript, Kim et al. employ mathematical models to analyse SARS-CoV-2 infection dynamics in the non-human primates in the presence or absence of remdesivir (a nucleotide analogue prodrug). The main model prediction is that early antiviral treatment with moderate efficacy (~30-70%) leads to a prolonged duration of viral shedding compared to infection duration without treatment. While the paper is well written, there are significant issues regarding the novelty and the technical details that would need to be addressed.

Reply to Reviewer summary:

Thank you very much for your thoughtful comments and suggestions, which helped us improve the quality of our manuscript. We have answered your questions and comments in a point-by-point manner as listed below:

Specific comments:

1. In their recent work in PLoS Biology (PMID: 33750978), the authors made similar predictions on SARS-CoV-2 dynamics during antiviral treatments: 'Note that viral shedding may last longer with treatment than without treatment if the antiviral efficacy is below 100% and initiated early.' Another modelling study (PMID: 33097472) from a different group predicted that early initiation of subpotent remdesivir treatment prolonged the infection duration. Furthermore, a recent modelling study (PMID: 32882638) analysed SARS-CoV-2 infection dynamics during remdesivir treatment in the non-human primates and predicted that remdesivir might lengthen SARS-CoV-2 infection. Therefore, the novelty of this study is unclear.

Reply to comment 1:

We have clarified the novelty of the present study. First, longer viral shedding under incomplete antiviral treatment was suggested previously with the use of theoretical approaches in our manuscript and PMID: 33097472. However, not many studies have confirmed this by analyzing SARS-CoV-2 viral load data under treatment with the viral dynamics model. Specifically, we quantitatively analyzed longitudinal viral load data from different specimens of SARS-CoV-2-infected rhesus macaques without and with RDV treatment and confirmed the paradoxically longer durations of viral shedding due to antiviral treatment.

The previous study (PMID: 32882638) analyzed similar, but not the same, data with viral dynamics models; however, that analysis had critical limitations. First, they assumed that RDV “decreases the infected cell death rate” but did not cite any references. As we described in our paper, RDV’s main mode of action is to inhibit infection. Moreover, it is obvious that if infected cells live longer, the virus remains in the host longer (thus regardless of the timing and dose, RDV increases the duration of viral shedding). Second, that author did not directly estimate treatment efficacy. Rather, (individually) estimated parameter values were compared between animals without and with RDV treatment. In other words, antiviral treatment impacts all parameters regardless of the mode of action of RDV. Although we performed a simulation in which we varied the efficacy as the previous author did, our presentation of the estimation of the efficacy is more informative. Otherwise, all simulations could be taken as surreal. Third, the statistical approach (Mann-Whitney U test) used in the other study is inappropriate because: 1) the Mann-Whitney U test is the correct test for examining differences in distributions (not the median or mean), and 2) testing the distribution of point estimations does not make sense.

Regardless of these critical limitations, the previous study is the first work suggesting the possibility of a negative side of RDV treatment. To respect their work, we cited the paper. Because we do not want to emphasize these critical limitations, we slightly changed the corresponding sentences in the Discussion section (Page 12, Line 239).

2. In their model (Eqs. [1-2]), when $\epsilon = 1$, the viral load decline is given by $dV/dt = -\delta \cdot V$. In the standard SARS-CoV-2 dynamics model, when $\epsilon = 1$, the viral load decline is given by $dV/dt = -c \cdot V$. Here, δ and c are the infected cell death rate constant and virion degradation rate constant, respectively. Perhaps the authors could clarify this potential discrepancy.
3. It appears the authors have made pseudo steady state assumption ($dV/dt = 0$) to arrive at Eqs. [1-2]. Drug treatment (e.g., remdesivir) destroys pretreatment pseudo steady state (given by $V = pI/c$) and establishes a new pseudo steady state (given by $V = p(1-\epsilon)I/c$), where I is the infected cells, and p is viral production rate per infected cell. It is unclear whether their model (Eqs. [1-2]) can capture the viral load changes caused by changes in pseudo steady states due to remdesivir treatment initiation. Regardless, it

would help if the authors could compare model predictions with and without pseudo steady assumption in the supplementary.

Reply to comment 2&3:

As the reviewer pointed out, if the antiviral effect on virus replication is extremely high (i.e., close to 100%), the quasi-steady state assumption (QSSA) is violated. However, as we demonstrated, the antiviral effect of RDV is estimated to be around 60% or less. Thus, we still can apply QSSA. To confirm this, we compared the two models (i.e., the full and the reduced model) as shown in Fig L1.

[Figure removed by editorial staff per authors' request]

The full model is described as follows:

$$\begin{aligned}\frac{dT(t)}{dt} &= -\beta T(t)V(t), \\ \frac{dI(t)}{dt} &= \beta T(t)V(t) - \delta I(t), \\ \frac{dV(t)}{dt} &= (1 - \varepsilon H(t))pI(t) - cV(t).\end{aligned}$$

Reasonable parameter values were used for illustration purposes: $\beta = 6.7 \times 10^{-6}$, $\delta = 1.14$, $p = 4.3 \times 10^6$, and $c = 10$. Note that the inhibition rate is 60% ($\varepsilon = 0.6$). The predictions of the full model (green curves) and the reduced

model (red curves) are similar. Further, we confirmed that RDV treatment (with 60% efficacy) resulted in paradoxically longer durations of viral shedding in both models.

[Figure removed by editorial staff per authors' request]

Further, we assessed the impact of QSSA on parameter estimation. Fig L2 shows the fitted curve of the full model (c is fixed as 10). The estimated fixed effect parameters by the two models are listed in Table L1. We found no big differences in estimated parameter values, which thus implies that QSSA can be applied to this dataset. To clarify these points, we updated the Discussion section (Page 12, Line 243).

Table L1. Comparison of estimated parameters

Parameter (Unit)	Reduced model		Original model	
	Nose	Throat	Nose	Throat
γ (day ⁻¹)	18.2	2.89 [#]	19.6	3.73 [#]
β ((copies/ml) ⁻¹ day ⁻¹)	1.79×10^{-6}	6.70×10^{-6}	2.45×10^{-6}	9.17×10^{-6}
δ (day ⁻¹)		1.14		1.18
ε (-)		0.618		0.616
$V(0)$ (copies/ml)		2.86×10^3		4.66×10^3

[#] Statistically different from nose (the Wald test).

- The models in this manuscript does not explicitly consider innate and adaptive immune responses. I appreciate that estimating parameter values of more complex models with immune components might not be possible with the current data set. However, the authors must investigate (at least qualitatively) whether subpotent remdesivir treatment leads to prolonged infection duration even if immune responses are explicitly considered. For instance, the innate immune response has been suggested to render target cells refractory to infection (e.g., PMID: 32558354, PMID: 22761567). Could this prevent the slower rate of infection during subpotent remdesivir treatment and reduce infection duration?

Reply to comment 4:

As the reviewer pointed out, we considered a virus dynamics model with innate immune response by interferons (IFNs). The following model has been used in previous studies [1]:

$$\frac{df(t)}{dt} = -\frac{1}{1 + \eta F(t)} \beta f(t) V(t),$$

$$\begin{aligned}\frac{dV(t)}{dt} &= \frac{(1 - \varepsilon H(t))}{1 + \eta F(t)} \gamma f(t) V(t) - \delta V(t), \\ \frac{dF(t)}{dt} &= sV(t) - \alpha F(t),\end{aligned}\tag{L1}$$

where $f(t) = T(t)/T(0)$ is the relative fraction of uninfected target cells at time t to those at time 0, $V(t)$ is the concentration of SARS-CoV-2, and $F(t)$ is the concentration of IFNs produced from infected cells. We assumed QSSA in Eq.(L1). IFN is assumed to be secreted proportional to the amount of virus at a

[Figure removed by editorial staff per authors' request]

rate of $s = 1.0 \times 10^{-5}$ and cleared at a rate of $\alpha = 0.4$. We also fixed $F(0) = 1$, $\eta = 0.1$, and $V(0) = 1$ without a loss of generality.

The best-fit curves of this model are shown in **Fig L3**. Although the values are slightly changed, we confirmed that RDV treatment prolonged the duration of viral shedding even with the immune response model (L1). We added this result to the Supplementary Appendix (**Fig. S5** and **Table S6**).

Minor comments:

5. It would help if the authors could describe how they treated the data points falling below the detection limit of viral during data fitting. Were these data points ignored or set to zero during fitting?

Reply to comment 5:

In their Nature publication (2020), BN Williamson et al. used "1" (copies/ml) for the detection limit. In their datasets, viral load decreases to its detection limit at t^* (i.e., $V(t^*) = 1$) even if the viral load is above the detection limit (i.e., $V(t) > 1$) at $t = t^* - 1$ and $t = t^* + 1$ (e.g., positive at day 1, negative at day 2, positive at day 3). In this case, we assumed there is an observation error on $V(t^*)$. We treated these datasets as censored data. In our main text, we explain how we addressed this censoring issue as follows (Page 15, Line 311):

"To handle data under the detection limit, the likelihood was constructed assuming that the data are within the interval from 0 to α , where α is the lowest observed viral load (above the detection limit) around the day when the viral load is censored [2]."

6. On lines 261, the authors mention remdesivir dosing as, '10mg/kg loading dose on 0.5 days after inoculation with SARS-CoV-2, followed by 5mg/kg daily.' Should the authors consider the efficacy to vary with time instead of constant efficacy (e.g., PMID: 33097472)?

Reply to comment 6:

The loading dose in animal studies is typically high and the dose does not go below the lower maintenance capacity [3]. Further, as was explained in a previous publication (PMID: 33097472), the concentrations of the drug's active nucleoside triphosphate (NTP) component observed within peripheral blood mononuclear cells (PBMCs) in nonhuman primates are maintained at high levels. Therefore, the constant efficacy of RDV in our study is a reasonable assumption.

7. On line 75-82, the authors mention, 'For example, in the field of virus dynamics (21, 22), it is observed that viral shedding may paradoxically last longer with treatment than without treatment if the antiviral efficacy is below 100% ... In a recent paper (23), these paradoxical phenomena were observed for the first time in a SARS-CoV-2 infection non-human primate model treated with RDV ...' It seems experiments in ref. (23) measure the viral load only up to day 7 post-infection. Therefore, it is unclear how the authors interpret and state that the infection duration is longer with remdesvir (RDV) than without treatment in these experiments.

Reply to comment 7:

Thank you for pointing this out. We updated the sentences as follows:

"In a recent paper (23), these paradoxical phenomena were observed for the first time in a SARS-CoV-2 infection non-human primate model treated with RDV, which is known as a nucleoside analogue."

8. References 24 and 38 are the same.

Reply to comment 8:

We removed Reference 38.

9. On line 131, is it reference 24 or is it 23?

Reply to comment 9:

We changed it.

1. Baccam P, Beauchemin C, Macken CA, Hayden FG, Perelson AS. Kinetics of influenza A virus infection in humans. *J Virol.* 2006;80(15):7590-9. Epub 2006/07/15. doi: 80/15/7590 [pii] 10.1128/JVI.01623-05. PubMed PMID: 16840338; PubMed Central PMCID: PMC1563736.
2. Adeline Samson, Marc Lavielle, France Mentré. Extension of the SAEM algorithm to left-censored data in nonlinear mixed-effects model: Application to HIV dynamics model. *Computational Statistics & Data Analysis.* 2006;51(3):1562-74. doi: 10.1016/j.csda.2006.05.007.
3. Roberts DM, Sevastos J, Carland JE, Stocker SL, Lea-Henry TN. Clinical Pharmacokinetics in Kidney Disease: Application to Rational Design of Dosing Regimens. *Clin J Am Soc Nephrol.* 2018;13(8):1254-63. Epub 2018/07/26. doi: 10.2215/CJN.05150418. PubMed PMID: 30042221; PubMed Central PMCID: PMC6086693.

July 19, 2021

RE: Life Science Alliance Manuscript #LSA-2021-01049R

Dr. Shingo Iwami
Department of Biology, Faculty of Sciences, Kyushu University
Motooka
Fukuoka 8190395
Japan

Dear Dr. Iwami,

Thank you for submitting your revised manuscript entitled "Incomplete antiviral treatment induce longer durations of viral shedding during SARS-CoV-2 infection". We would be happy to publish your paper in Life Science Alliance pending final revisions necessary to meet our formatting guidelines. Please also incorporate the remaining minor edits suggested by the Reviewer.

- please note that titles in the system and manuscript file must match
- please mark your 2ndary corresponding author (Keisuke Ejima) in the system as well
- please consult our manuscript preparation guidelines <https://www.life-science-alliance.org/manuscript-prep> and make sure your manuscript sections are in the correct order
- please rename the "Results and Discussion" section to "Results" and the "Conclusions" section to "Discussion"
- we encourage you to revise the figure legends for figures 1, S2, S3, S4, S5 such that the figure panels are introduced in alphabetical order
- please add your main, supplementary figure, and table legends to the main manuscript text after the references section
- please add callouts for all panels in Supplementary figures to your main manuscript text
- please remove references from the Supplemental Figures and incorporate them instead into the main Reference list

LSA now encourages authors to provide a 30-60 second video where the study is briefly explained. We will use these videos on social media to promote the published paper and the presenting author. Corresponding or first-authors are welcome to submit the video. Please submit only one video per manuscript. The video can be emailed to contact@life-science-alliance.org

A. FINAL FILES:

B. MANUSCRIPT ORGANIZATION AND FORMATTING:

Sincerely,

Reviewer #1 (Comments to the Authors (Required)):

The authors have addressed my concerns satisfactorily. Therefore, I only have some minor comments/suggestions (please see below.)

- On line 27, could the authors consider using 'some mathematical models' or 'a class of mathematical models' or something similar instead of 'mathematical theory'? My understanding is that only some models predict early antiviral treatment with moderate efficacy leads to a prolonged duration of viral shedding. The prediction of course depends on the model assumptions.

- On lines 77-79, the authors mention, 'For example, in the field of virus dynamics, it is observed ...' This sentence appears to suggest all viral dynamics models make these paradoxical predictions. Here again, the authors could consider toning down their statement.

- On lines 87 and 166, the authors mention '... to better understand the mechanism ...' and 'To elucidate a mechanism ...' Their model certainly provides insight into the infection dynamics during treatment. However, it is not clear if the model is providing mechanistic insights.

- Either in the introduction or discussion, the authors could briefly describe contributions of mathematical models to our understanding of SARS-CoV-2 infection dynamics and treatments (e.g., PMIDs: 33097472, 34228712, 33750978, 33536313, 33290397). This may provide a good context for their model and work.

Response to the reviewers' comments

Kwang Su Kim, Shoya Iwanami, Takafumi Oda, Yashuhisa Fujita, Keiji Kuba,
Taiga Miyazaki, Keisuke Ejima and Shingo Iwami

July 22, 2021

Reviewer #1 (Comments to the Authors (Required)):

The authors have addressed my concerns satisfactorily. Therefore, I only have some minor comments/suggestions (please see below.)

Minor comments:

1. In On line 27, could the authors consider using 'some mathematical models' or 'a class of mathematical models' or something similar instead of 'mathematical theory'? My understanding is that only some models predict early antiviral treatment with moderate efficacy leads to a prolonged duration of viral shedding. The prediction of course depends on the model assumptions.

Reply to comment 1:

Thank you for comments. We changed it.

2. In On lines 77-79, the authors mention, 'For example, in the field of virus dynamics, it is observed ...' This sentence appears to suggest all viral dynamics models make these paradoxical predictions. Here again, the authors could consider toning down their statement.

Reply to comment 2:

As the reviewer commented, we modified this sentence.

3. On lines 87 and 166, the authors mention '... to better understand the mechanism ...' and 'To elucidate a mechanism ...' Their model certainly provides insight into the infection dynamics during treatment. However, it is not clear if the model is providing mechanistic insights.

Reply to comment 3:

As the reviewer commented, we modified this sentence.

4. Either in the introduction or discussion, the authors could briefly describe contributions of mathematical models to our understanding of SARS-CoV-2 infection dynamics and treatments (e.g., PMIDs: 33097472, 34228712, 33750978, 33536313, 33290397). This may provide a good context for their model and work.

Reply to comment 4:

We would like to thank the reviewer's helpful comment. We added sentence and references in the revised manuscript.

July 23, 2021

RE: Life Science Alliance Manuscript #LSA-2021-01049RR

Dr. Shingo Iwami
Department of Biology, Faculty of Sciences, Kyushu University
Motooka
Fukuoka 8190395
Japan

Dear Dr. Iwami,

Thank you for submitting your Research Article entitled "Incomplete viral treatment may induce longer durations of viral shedding during SARS-CoV-2 infection". It is a pleasure to let you know that your manuscript is now accepted for publication in Life Science Alliance. Congratulations on this interesting work.

*****IMPORTANT:** If you will be unreachable at any time, please provide us with the email address of an alternate author. Failure to respond to routine queries may lead to unavoidable delays in publication.*******

DISTRIBUTION OF MATERIALS:

Again, congratulations on a very nice paper. I hope you found the review process to be constructive and are pleased with how the manuscript was handled editorially. We look forward to future exciting submissions from your lab.

Sincerely,
